# Analysis of the Microstructure and Mechanical Properties of TiB_w_/Ti-6Al-4V Ti Matrix Composite Joint Fabricated Using TiCuNiZr Amorphous Brazing Filler Metal

**DOI:** 10.3390/ma14040875

**Published:** 2021-02-12

**Authors:** Hao Tian, Jianchao He, Jinbao Hou, Yanlong Lv

**Affiliations:** 1Aeronautical Key Laboratory for Welding and Joining Technologies, AVIC Manufacturing Technology of Institute, Beijing 10024, China; tianhao19961207@163.com (H.T.); houjb@avic.com (J.H.); lvy006@avic.com (Y.L.); 2Institute of Speical Environments Physical Sciences, Harbin Institute of Technology, Shenzhen 518055, China

**Keywords:** ceramic-reinforced Ti matrix composite (TMC), in situ synthesized TiB whisker, TiB_w_/Ti-6Al-4V titanium matrix composite, high-temperature tensile property

## Abstract

TiB crystal whiskers (TiB_w_) can be synthesized in situ in Ti alloy matrix through powder metallurgy for the preparation of a new type of ceramic fiber-reinforced Ti matrix composite (TMC) TiB_w_/Ti-6Al-4V. In the TiB_w_/Ti-6Al-4V TMC, the reinforced phase/matrix interface is clean and has superior comprehensive mechanical properties, but its machinability is degraded. Hence, the bonding of reliable materials is important. To further optimize the TiB_w_/Ti-6Al-4V brazing technology and determine the relationship between the microstructure and tensile property of the brazed joint, results demonstrate that the elements of brazing filler metal are under sufficient and uniform diffusion, the microstructure is the typical Widmanstätten structure, and fine granular compounds in β phase are observed. The average tensile strength of the brazing specimen is 998 MPa under room temperature, which is 97.3% of that of the base metal. During the high-temperature (400 °C) tensile process, a fracture occurred at the base metal of the highest tensile test specimen with strength reaching 689 MPa, and the tensile fracture involved a combination of intergranular and transgranular modes at both room temperature and 400 °C. The fracture surface has dimples, secondary cracks are generated by the fracture of TiB whiskers, and large holes form when whole TiB whiskers are removed. The proposed algorithm provides evidence for promoting the application of TiB_w_/Ti-6Al-4V TMCs in practical production.

## 1. Introduction

The reinforcement of titanium matrix composites (TMCs) with ceramic phase results in superior wear resistance and specific strength [1,2,3]. Ti-6Al-4V reinforced using in situ synthesized TiB whisker is a new type of TMC. TiB_w_/Ti-6Al-4V is prepared using TiB_2_ as boron source through powder metallurgy. In comparison with the traditional TMCs reinforced by ceramic particles and fibers, the bonding performance at the interface between reinforced phase TiB and matrix is better in TiB_w_/Ti-6Al-4V [4], and the mechanical properties are not anisotropic [5]. High-performance TiB_w_/Ti-6Al-4V can be used as a substitute for some existing Ti alloys in various fields, such as aerospace, communication technology, and precision instrument manufacturing [6].

However, the addition of ceramic phase with extremely high hardness degrades the plasticity of TMCs and elevates the requirements for preparation conditions; consequently, it complicates the one-time formation of this kind of material, thus restricting the application of TMCs in the manufacturing of large complex structures [7].

Brazing is featured by overall heating and minor melting or un-melting of the matrix, thus, it can mitigate the adverse effects on the microstructural and mechanical properties of the matrix. Song et al. [8] studied the brazing of TiB_w_/Ti-6Al-4V to Ti60, but they only tested the shear strength of the joints. Considering that TiB has stable high-temperature thermodynamic properties and will not experience chemical reaction with fusion brazing filler metal, the research on Ti-6Al-4V brazing will be of a certain reference value to TiB_w_/Ti-6Al-4V. During the brazing process of Ti-6Al-4V, Pang et al. [9], Ba et al. [10], Shi et al. [11], Galindo [12], and Jing et al. [13] used Zr-, Cu-, and Ni-containing brazing filler metals as interlayers. Zr is completely miscible with Ti, thus exerting the solution a strengthening effect and improving joint strength hardness [14]. As stable elements of β-Ti, Cu, and Ni are able to lower the phase transition temperature of α-β in Ti-6Al-4V, causing transition from α-Ti to β-Ti at high brazing temperature. In the subsequent cooling process, β-Ti is decomposed into α-Ti and intermetallic compounds due to eutectoid reaction. This process causes tiny granular intermetallic compounds in diffusion zone of brazed joints, improving the strength and hardness of this zone to some extent.

In this study, TiBw/Ti-6Al-4V TMCs were bonded by vacuum brazing using Ti-Cu-Ni-Zr brazing filler. The microstructure and elemental distribution in the brazed joint were observed via scanning electron microscopy (SEM) and energy dispersive spectroscopy (EDS), and the tensile properties of the brazing specimens at room temperature and 400 °C were tested to reveal the corresponding relationship between the microstructure and tensile properties of TiBw/Ti-6Al-4V brazed joint.

## 2. Experiments

The TiB_w_/Ti-6Al-4V used in this study was prepared using TiB_2_ and Ti-6Al-4V mixed powder through low-energy ball milling and selective laser melting. The volume fraction of TiB whiskers was approximately 2 vol.%, and the morphology of its microstructure is shown in Figure 1a. The aspect ratio of TiB whiskers was approximately 5–15, mainly under reticular distribution at Ti-6Al-4V grain boundary, and a very few slenderer TiB whiskers were inside grains. The grain size of Ti-6Al-4V was approximately 170 μm × 165 μm with equiaxed morphology, and the intragranular bright needle-like phase was β-Ti. The TiB_w_/Ti-6Al-4V was cut into 30 × 30 × 30 mm^3^ and 20 × 20 × 30 mm^3^ specimens by using a linear cutting device. Ti-Cu-Ni-Zr amorphous foil strip with thickness of 20 μm was prepared through vacuum melt-spinning technique and was taken as the interlayer. The nominal component was Ti-10Cu-10Ni-5Zr, and the actual components are listed in Table 1. Before brazing, 320–800 mesh sand paper was used to grind the specimen surface, followed by ultrasonic cleaning in alcohol. The Ti-Cu-Ni-Zr foil strip was placed between the specimens, and a pressure of 5 MPa was applied above them for brazing. Then, the specimen was cut into 10 × 2 × 2 mm^3^ metallurgical specimens and then subjected to a tensile test to observe the joint microstructure and tensile properties at room and high temperature through the tensile test.

According to Jing et al. [13], when Ti-Cu-Ni-Zr brazing filler metal was used for Ti-6Al-4V brazing, the result was enhanced at a brazing temperature of over 900 °C. The liquidus temperature of the filler metal was around 920 °C, and the β-phase transition temperature of Ti-6Al-4V was approximately 995 °C. Coarse grains caused by the α–β phase transition of the base metal can be avoided by keeping the brazing temperature below this temperature. Meanwhile, lengthening the holding time can homogenize the elemental diffusion of the brazing filler metal to avoid the formation of hard and brittle compounds in the brazing seam. Hence, the brazing parameters selected in the test were as follows: 940 °C and holding time of 2 h; the concrete brazing process is shown in Figure 1b. The brazing process was implemented in a vacuum brazing furnace with vacuum degree of 2.0 × 10^−3^ Pa. The tensile test was performed on a universal testing machine at rate of 0.5 mm/min, and the average values of tensile strength in the three groups of parallel specimens were obtained in each group. The microstructure and tensile fracture morphology of TiB_w_/Ti-6Al-4V brazed joint were observed using a S-3400 SEM device HITACHI, Tokyo, Japan), and the elemental composition was analyzed using EDS (HITACHI, Tokyo, Japan).

## 3. Results and Discussion

### 3.1. Microstructure of TiB_w_/Ti-6Al-4V Brazed Joint

The microstructure and elemental diffusion of the TiB_w_/Ti-6Al-4V brazed joint were analyzed in this section. Figure 2a,b shows the SEM images of TiBw/Ti-6Al-4V joint brazed using Ti-Cu-Ni-Zr amorphous brazing filler metal at brazing temperature of 940 °C for 2 h. The images show that no micro-crack and pore appeared in the joints. Based on pervious results obtained by other scholars [12,13,14]; normally, the brazed joints include two parts: fusion zone (FZ) and diffusion zone (DZ). However, no evident boundary was observed between this two zones in this study. This phenomenon should benefit from sufficient elemental diffusion of the brazing filler metal. Generally, the boundary of the brazing diffusion zone is flat and straight, indicating that neither intergranular nor intragranular TiB whiskers impeded the elemental diffusion. Figure 2c shows the EDS line scanning results of the joint zone. The diffusion distance of Cu was largely identical with that of Ni, and the size of the whole brazing joint zone was approximately 200 μm. Moreover, in comparison with the base metal that did not undergo high-temperature thermal cycle in brazing process, the grain size of the base metal of the brazing specimen was not coarsened, and TiB whiskers were still mainly distributed at the Ti-6Al-4V grain boundary in reticular structure, mainly because the TiB whiskers at the grain boundary inhibited the coarsening phenomenon of Ti-6Al-4V grains at high temperature.

The brazing joint mainly included bright needle-like phase (marked as A) under lamellar distribution, dark needle-like phase (marked as B), and rod-like phase with flat and straight interface (marked as C). The EDS analysis results of points A, B, and C are shown in Table 2. The elements at points A and B comprised mainly Ti. Relatively, the Cu and Ni contents were high in A, while the Al content was higher in B. Cu and Ni were β-phase stabilized elements, and Al could serve as an α-phase stabilized element. Hence, the dark needle-like phase marked by point B should be the α-Ti phase realizing the solid solution of a small quantity of brazing filler metal elements. Based on the magnified micrograph shown in Figure 2b, the bright needle-like phase marked by point A did not have a single-phase microstructure, but rather centered on the β-Ti phase, in which fine particle phases were present. The concrete phase composition in A will be discussed in detail. Notably, unlike the reticular α-β dual phase structures in the base metal, because the generated of the bright needle-like phase in DZ is dominated by the diffusion of Cu and Ni in brazing joint, the Widmanstätten structure in DZ was parallel, its growth direction was consistent with the elemental diffusion direction in the brazing joint. The mass fractions of Ti and B in point C were approximately 61.9 and 35.5, respectively, manifesting that this phase was a reinforced TiB phase.

The dotted block in Figure 2b was amplified to obtain the abovementioned magnified SE image of lamellar structure. Both Wang et al. [15] and Xia et al. [16] found similar morphologies when using Ti-Zr-Cu-Ni brazing filler metal to braze Ti-6Al-4V. According to these results, this lamellar structure represent the Widmanstätten structure generated by eutectoid reaction, and its formation mechanism could be explained as follows: When Cu and Ni, which are β-phase stabilized elements in brazing filler metal, were diffused into the TiB_w_/Ti-6Al-4V base metal, the α–β phase transition temperature of Ti alloy will decrease, resulting in the phase transition of the base metal in the diffusion zone at high brazing temperature and β–Ti phase with the solid solution of Cu and Ni. In the subsequent cooling process, the nucleation of α-phase occurred, and the α-Ti grew up to form lamellar structure. As the temperature was further lowered, the granular compounds were precipitated from needle-like β-Ti phase and diffused around the original needle-like β-Ti in granular morphology, resulting in the final microstructure of α-Ti + β-Ti+ granular compounds. Meanwhile, this eutectoid reaction further refined α phase in DZ. In addition, different from α phase with disorderly orientation in the base metal, the growth direction of the α phase in the joint zone remained consistent in a direction that is approximately perpendicular to the central line of the brazing seam, indicating that the morphology of Widmanstätten structure depended on the diffusion of Zr, Ni, and Cu in the brazing filler metal.

To study the concrete composition of granular compounds, we implemented the EDS surface scanning of bright needle-like phase under magnified SEM image, and the results are displayed in Figure 2d. The EDS maps show that Cu and Ni were aggregated in the bright needle-like phase. Jing et al. [13,14] once explored the effect of Ti-Zr-Cu-Ni brazing filler metal on Zr during Ti-6Al-4V brazing. Jing believed during brazing, Zr can be dissolved into α and β-Ti phases to exert a solution-strengthening effect or form Ti-Zr-Cu-Ni compounds when the Zr content in the brazing filler metal is high enough (>18%). Generally, (Ti,Zr)_2_(Cu,Ni) is the concrete elemental composition of granular compounds, but the formation mechanism remains unclear. In terms of standard Gibbs formation energy, Ti could react with Cu and Ni more easily than Zr. He et al. [17] and Kundu et al. [18] probed Ti–Cu and Ti–Ni reaction products during brazing process in terms of thermodynamics. According to their calculation results, Ti_2_Cu and TiNi_3_ can be easily generated under Ti–Cu and Ti–Ni binary systems, and Zr appeared in the compounds through replacement reaction.

### 3.2. Mechanical Properties of TiBw/Ti-6Al-4V Brazed Joint

The tensile properties of the TiB_w_/Ti-6Al-4V brazed joint at room temperature and 400 °C were tested in this section, the fracture surfaces of the tensile test specimens were observed, and the occurrence reason for fracture was explored. Table 3 shows the high-temperature tensile test results of TiB_w_/Ti-6Al-4V base metal and those of TiB_w_/Ti-6Al-4V brazed joint obtained through brazing at 940 °C (held for 2 h). Under the test parameters in this study, the fracture of the tensile test specimen occurred in the diffusion zone at room temperature, where the average tensile strength was 998 ± 20 MPa, reaching 97.3% of that of the original base metal. The fracture of TiB_w_/Ti-6Al-4V tensile test specimen occurred in the base metal at 400 °C, and the average tensile strength was 679 ± 25 MPa, which slightly declined compared with that of the original base metal.

Figure 3 shows the SEM images of the tensile fracture surfaces of TiB_w_/Ti-6Al-4V brazing specimen at room temperature and 400 °C. Considering that the TiB_w_/Ti-6Al-4V tensile test specimens were tested under room and high temperature, although the cracking position varied, the morphology of fracture surfaces was similar and mainly involved intergranular fracture, while individual grains underwent transgranular fracture. Furthermore, based on the magnified image, the room-temperature tensile fracture of the brazing test specimen involved the combination of intergranular and transgranular modes, and the surface contained many granular projections and fine holes distributed in between, and cleavage plane with smooth vertical and horizontal distribution surface and large holes formed by the de-bonding of TiB/matrix interface. According to the analysis results of joint microstructure in Section 3.1, the granular projections in the fracture were granular compounds adhered to the β phase in the diffusion zone. During the tensile process, the bonding interfaces of granular compounds with α and β phases were de-bonded, thus forming fine pores, which were aggregated and expanded to form cracks. Relatively, the high-temperature fracture morphology of the TiB_w_/Ti-6Al-4V brazing tensile test specimen also had the three above features with the following differences were: (1) the surface nearby the fine holes in the high-temperature fracture was smoother, and these fine holes were dimples generated by a typical microvoid coalescence-type fracture; (2) at high temperature, the proportion of TiB whiskers that were fractured or pulled out was larger, and evident secondary cracks appeared on the fracture surface. Hence, three fracture mechanisms simultaneously existed in the tensile test at either or high temperature. However, the proportions of the three fracture forms might vary with test conditions and fracture position.

As the temperature increased, the plasticity of Ti-6Al-4V was improved. Hence, the cracking and pullout of TiB whiskers during the high-temperature tensile process were caused by the plasticity difference between Ti-6Al-4V and reinforced phase TiB at high temperature. In the tensile process at 400 °C, a larger deformation of Ti-6Al-4V matrix was formed because of enhanced plasticity. Consequently, the TiB whiskers distributed at the grain boundary bore larger 3D stresses. Finally, they were cracked or pulled out on the whole. Hence, the reinforced phase TiB was not an obstruction for crack elongation but the original position of cracking at high temperature. The high-temperature fracture that occurred in the base metal also verified this conclusion. The tensile strength of the base metal after brazing at 400 °C was slightly lower than that of the original base metal, indicating that the thermal cycle of brazing damaged the high-temperature performance of the base metal, and the main cause needs further investigation.

## 4. Conclusions

To explore the corresponding between the microstructure and tensile properties of TiBW/Ti-6Al-4V brazed joint and optimize its mechanical property data, we used a 200 nm-thick Ti-Cu-Ni-Zr amorphous brazing foil strip to perform the brazing test of TiBW/Ti-6Al-4V TMCs at 940 °C (held for 2 h), and the results were as follows:

The morphology of TiB_w_/Ti-6Al-4V TMC joint in the fusion zone did not evidently differ from that in the diffusion zone, where Widmanstätten structures were attributed to the base metal structure, and the main phase compositions included lamellar parallel α-Ti phase and β-Ti phase, and fine granular compounds.The influence of elemental diffusion in the brazing filler metal on the microstructure was mainly manifested by the process refined the α phase, while the growth direction of the lamellar structure was turned from the reticular type of the base metal into the parallel type, which is basically consistent with the diffusion direction. The TiB whiskers distributed at the grain boundary almost did not exert any influence on the elemental diffusion in the brazing filler metal, and the boundary in the diffusion zone was flat and straight.The tensile strength of the TiBw/Ti-6Al-4V TMC brazing specimens was 998 MPa at room temperature, reaching 97.3% of the original base metal. The tensile fracture occurred in the base metal at 400 °C and the strength was 679 MPa. The specimen fracture in the tensile test at room temperature differed from that at high temperature. At room temperature, TiB whiskers impeded the crack expansion and consumed the energy of cracking elongation by changing their direction or through the fracture of their own. However, considering the plasticity difference of Ti-6Al-4V matrix at high temperature, TiB whiskers bore three-dimensional stresses generated by mass deformations, resulting in fracture or removal.

## Figures and Tables

**Figure 1 materials-14-00875-f001:**
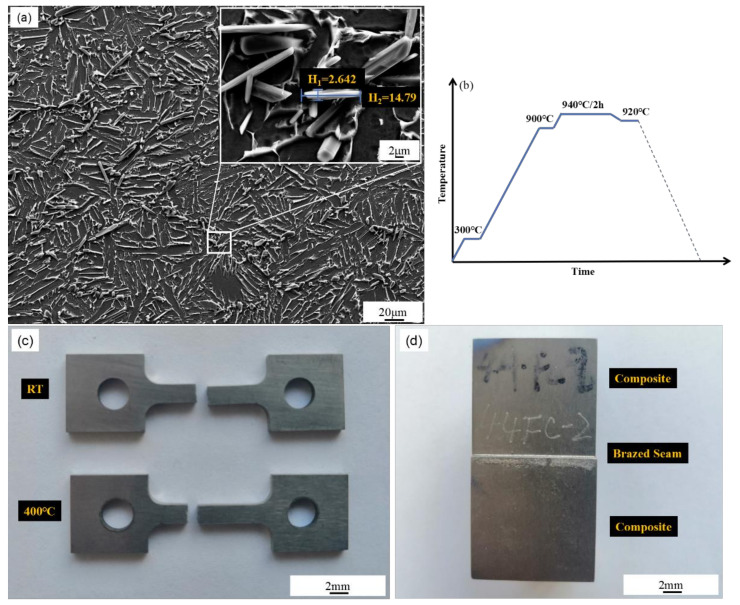
(**a**) Microstructure of TiB_w_/Ti-6Al-4V composite; (**b**) time–temperature heating curve at brazing process; (**c**) image of tensile-test samples, and (**d**) brazing sample.

**Figure 2 materials-14-00875-f002:**
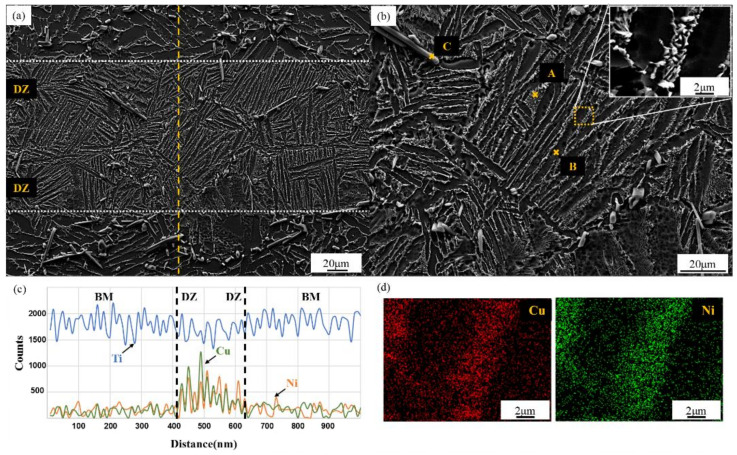
Microstructure and elemental distribution of the TiB_w_/Ti-6Al-4V composite joint brazed with Ti-Cu-Ni-Zr filler metal at 940 °C for 2 h: (**a**) the whole microstructure of the joint; (**b**) a magnified view of the brazing seam; (**c**) line scanning results corresponding to line marked in (**a**); (**d**) elemental distribution maps corresponding to the magnified view of Widmanstätten in (**b**).

**Figure 3 materials-14-00875-f003:**
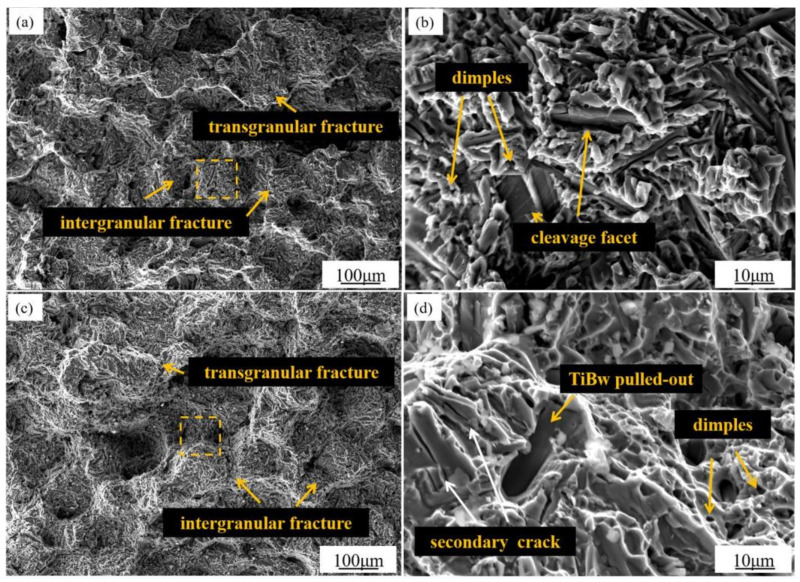
Fracture surface of TiB_w_/Ti-6Al-4V brazing joints brazed in 940 °C for 2 h (**a**,**b**) room-temperature; (**c**,**d**) 400 °C.

**Table 1 materials-14-00875-t001:** Chemical composition of TiBw/Ti-6Al-4V composite and filler metal (wt.%).

Test Items	Ti	Al	V	Cu	Ni	Zr	B
TiB_w_/Ti-6Al-4V	Bal.	6.7	4.2	-	-	-	2.3
Filler metal	Bal.	-	-	11.3	9.5	4.9	-

**Table 2 materials-14-00875-t002:** Chemical composition (wt.%) and possible phase of A, B, C spots marked in Figure 2b.

Spot	Ti	Cu	Ni	Al	Zr	V	B	Possible phase
A	73.6	7.8	7.3	3.3	4.4	2.6	-	β-Ti + undetermined compounds
B	88.6	1.3	0.3	6.0	2.7	1.0	-	α-Ti
C	61.9	0.5	0.3	0.3	0.6	0.9	35.5	TiB

**Table 3 materials-14-00875-t003:** Result of tensile strength test at room-temperature and 400 °C (MPa).

Test Items	Room-Temperature	400 °C
Brazing joints	998 ± 20	679 ± 25
TiB_w_/Ti-6Al-4V	1020 ± 5	747 ± 25

## Data Availability

The data presented in this study are available on request from the corresponding author.

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
