# Peer review of "Analysis of the Microstructure and Mechanical Properties of TiBw/Ti-6Al-4V Ti Matrix Composite Joint Fabricated Using TiCuNiZr Amorphous Brazing Filler Metal"

_materials, 2021, doi:10.3390/ma14040875_

Round 1

Reviewer 1 Report

The authors improved the strength and bonding strength of the Ti alloy through the TiB whisker. All contents are well designed and structured with logic, but the authors need to add some data to support conclusion. In the tensile test, the number of samples tested and the error bar of the data must be added. There is an error in the analysis tool. please correct SE image to SEM image.

Author Response

Thank you for your precious comments. The following is my responds to your questions:

Reviewer 2 Report

Review

The article requires revision.

There are a few notes:

Line 18

Pascal is indicated with a small letter.

Lines 18 and 21.

It is necessary to indicate the error when measuring the tensile strength.

Line 33

Abbreviation used «powder metallurgy» - «PM». This abbreviation is not used further in the text.

Lines 52-54

«Cu and Ni can act as β-phase stable elements, lower the α–β phase transition temperature of Ti-6Al-4V, elevate the β-phase content in the joint, and exert joint-strengthening effect.»

The statement that Cu and Ni increase the amount of the beta phase in the compound and this will lead to the hardening effect is incorrect! Cu and Ni lead to the formation of the beta phase only at a high temperature, that is, at the brazing temperature, but when the brazed seam is cooled, decomposition occurs according to the eutectoid mechanism. Authors, look at the alloy state diagrams! The beta phase, stabilized by Cu and Ni, decomposes into alpha Ti and intermetallic compounds, which leads to a slight increase in strength and a strong decrease in ductility. It is described in the literature that the decomposition kinetics is so intense that there are no cooling rates at which the beta titanium phase stabilized by Cu and Ni can be fixed at room temperature.

Line 65

Figure 1a shows the particle size «H1» и «H2». Size H1 has 3 decimal places, H2 size has 2 decimal places. The quality of the picture needs to be improved. It is also recommended to split this figure into several, as it shows both the microstructure and the soldering mode and the external views of the images. Everything is mixed!

Line 70

The statement about the use of amorphous foil 200 nanometers thick raises very serious doubts. Typically, the thickness of such solders is 40-60 microns.

Line 73

The author writes «Before welding». Probably need to write brazing.

Line 75

Pressure is not measured in kilonewtons (кН).

Lines 73-78

The authors describe the operations of sample preparation and testing. Not a word was said about the brazing mode itself (temperature, time). There is not even a link to figure 1, which shows the heating curve. It is also not clear what caused such a regime. The work does not show the melting point of the solder (at least from the literature).

Lines 78-79

Image 1d says "Filler metal" in one of the areas, but this is not "filler metal", but "brazed seam"

Figure 2 introduces the designations "DZ" and "FZ", which are not described in the text.

Line 102

The author again confuses the concepts of welding and brazing. Welding is a completely different process.

Lines 126-139

The description of the formation of the structure should be revised taking into account the comments to lines 52-54.

Line 141

The phase assumption at point "A" is incorrect.

Lines 184 and 186

Error of the tensile strength is not specified.

Line 188

Table name and table are on different pages.

Table 3

The error is not specified. The measurement accuracy of 0.x MPa is also questionable. Moreover, all values must be recorded in the same way with the same measurement accuracy..

Conclusions

It is necessary to correct taking into account the comments

Author Response

(The authors gave the same response as above.)

Reviewer 3 Report

1) Even if it is a field standard, please provide an initial designation for the abbreviation TiBw.

2) Too much of the first paragraph of your Experiments section contains results from the powder preparation including grain size. Please move to the Results and Discussion section 

3) Your paper states that the filler metal used is 200 nm thick, yet your diffusion and fusion zones are well over 100 microns thick. Please provide a cross-sectional image showing that your filler metal is indeed nearly 1,000 times thinner than your diffusion and fusion zones

4) Please double-check your manuscript for grammar and spelling errors

5) I would hesitate to call the fracture "mainly intergranular" since it only appears to be the case for the TiBw phases. The matrix seems to be another story

Author Response

(The authors gave the same response as above.)

Round 2

Reviewer 2 Report

The remarks have been eliminated. The article can be published

Reviewer 3 Report

All concerns have been adequately addressed